# A Comprehensive Study on Tear Meniscus Height Inter-Eye Differences in Aqueous Deficient Dry Eye Diagnosis

**DOI:** 10.3390/jcm13030659

**Published:** 2024-01-23

**Authors:** Hugo Pena-Verdeal, Jacobo Garcia-Queiruga, Belen Sabucedo-Villamarin, Carlos Garcia-Resua, Maria J. Giraldez, Eva Yebra-Pimentel

**Affiliations:** 1GI-2092—Optometría, Departamento de Física Aplicada (Área de Optometría), Facultade de Óptica e Optometría, Universidade de Santiago de Compostela, Campus Vida S/N, 15705 Santiago de Compostela, Spain; belen.sabucedo@rai.usc.es (B.S.-V.); carlos.garcia.resua@usc.es (C.G.-R.); mjesus.giraldez@usc.es (M.J.G.); eva.yebra-pimentel@usc.es (E.Y.-P.); 2Instituto de Investigación Sanitaria (IDIS), Travesía da Choupana S/N, 15706 Santiago de Compostela, Spain

**Keywords:** dry eye disease, tear meniscus height, aqueous deficient dry eye, diagnostic criteria, inter-eye differences

## Abstract

(1) Background: Dry eye disease (DED) is a chronic ocular surface condition that requires precise diagnostic tools. The present study aimed to investigate the diagnostic potential of the absolute inter-eye difference (|OD-OS|) in tear meniscus height (TMH) for the detection of the presence of aqueous deficient dry eye (ADDE). (2) Methods: A sample of 260 participants with dry eye complaints underwent ocular surface examinations thorough diagnostic assessments based on the Tear Film and Ocular Surface Society guidelines (TFOS DEWS II). Participants were subsequently categorized as No ADDE and ADDE based on TMH. Statistical analyses to determine the optimal TMH|OD-OS| cut-off value in a randomly selected study group (200 participants) were performed, while a separate validation analysis of the cut-off value obtained in a random cross-validation group (60 participants) was also performed. (3) Results: The significant diagnostic capability of TMH|OD-OS| (area under the curve = 0.719 ± 0.036, *p* < 0.001) was found. The identified cut-off value of 0.033 mm demonstrated reliable specificity (77.6%) and moderate sensitivity (59.1%). Cross-validation confirmed the cut-off value’s association with the TFOS DEWS II diagnostic criterion (Cramer’s V = 0.354, *p* = 0.006). (4) Conclusions: The present study provides evidence for the diagnostic potential of TMH|OD-OS| in identifying ADDE. The identified cut-off value enhances the specificity and offers moderate sensitivity, providing an objective tool for clinical decision making.

## 1. Introduction

Dry eye disease (DED) underwent a redefinition by the Tear Film and Ocular Surface Society (TFOS) during the second Dry Eye Workshop (DEWS II) [1]. In its first edition, the TFOS defined the condition as a multifactorial tear film and ocular surface disease that leads to ocular symptoms associated with tear film instability, hyperosmolarity, and ocular surface damage and inflammation [2]. Since DED could originate from various factors, the new definition proposed in the DEWS II report focuses on the multifactorial nature of the disease, with the disruption of tear film homeostasis as the key factor, accompanied by ocular symptoms. The etiological role is played by the factors defined in the previous report, with the addition of neurosensory abnormalities to the definition [1,3,4]. All these factors cause the patient to fall into the vicious circle of DED [4,5].

The main hypothesis proposes that a patient’s entry into the vicious circle of DED is associated with dissecting stress, primarily influenced by reduced tear production or an increased tear evaporation rate [4]. In both situations, homeostasis is disturbed, leading to hyperosmolarity at the ocular surface, which subsequently triggers the activation of the inflammatory cascade, ultimately destabilizing the tear film [6,7]. The elevation in protein and electrolyte concentrations, resulting from a decrease in tear volume that initially provokes irritation on the ocular surface, progresses to trigger inflammation and subsequent harm in evaporative dry eye. The thinning of the lipid layer facilitates heightened evaporation [6,7]. Subsequently, hyperosmolarity induces apoptosis, acting as pro-inflammatory stress and diminishing the capacity of mucin-like molecules to provide lubrication to the ocular surface, leading to potential permanent damage [8]. Once this occurs, various compensatory events take place, such as increased tear production or an increased blinking rate, and patients begin to complain of symptoms [4]. As the loss of homeostasis is the main factor in the definition, this leads clinicians to pursue strategies that attempt to restore normal ocular function by paying attention to all possible entry factors into the vicious circle of disease. Furthermore, researchers emphasize the necessity for additional investigations into procedures aimed at pinpointing the DED diagnosis and accurately identifying its various types.

Since there are different etiological factors, two primary DED types have been identified in the TFOS DEWS II Diagnostic Methodology report: aqueous deficient dry eye (ADDE) and evaporative dry eye (EDE) [1]. The tear aqueous component plays a critical role in maintaining ocular surface health, and its dysregulation serves as both a crucial pathogenic mechanism and a diagnostic sign in the ADDE subtype. The tear volume and production are pivotal components of the tear dynamics, directly associated with the final total aqueous component. A reduction in tear volume is recognized as a diagnostic indicator of the presence of the ADDE subtype [9,10]. The assessment of the tear meniscus, modulated by the tear volume and production rate, serves as a widely employed primary diagnostic tool for the discrimination of individuals with ADDE [3,11]. Regarding the EDE subtype, the homeostasis alteration is caused by compromised tear film stability that does not adequately protect the ocular surface [4]. Tear film stability highly depends on the adequate function of the lipid layer, whose primary component is the meibum secreted by the meibomian glands [4]. An alteration in these glands may lead to blepharitis and an inefficient lipid layer, causing an enhancement in tear film evaporation. Meibomian gland dysfunction (MGD) may be determined by observing the morphological structure of the glands and assessing their appearance and function on the eyelid margin, where the orifices of the glands are located and the meibum is secreted [12]. Signs of EDE and ADDE may overlap in the most severe cases of DED, which are highly prevalent in women and those of advanced age [13,14,15]. The meibomian glands suffer from various age-related changes, including meibomian gland atrophy and dropout and decreased acinar cell proliferation [13,15]. Moreover, a reduction in tear production seems to occur later in life, close to 50 years old [14]. Overall, patients may show signs that correspond to both subtypes, generating a subtype known as mixed dry eye. While these subtypes are defined when DED occurs, it is important to distinguish this condition from other diseases or conditions affecting the ocular surface. This differentiation is essential, especially in cases where symptoms are present without corresponding signs or when signs are evident without accompanying symptoms [1]. On one hand, patients could be exhibiting symptoms with no signs, leading to the diagnosis of a pre-clinical state of DED or neuropathic pain. On the other hand, the presence of signs without symptoms could be linked to a predisposition to a DED state or a neurotrophic condition [1]. These situations should be clearly identified to eliminate confounding factors that may impact the management of DED.

There are different procedures to assess the functionality of the tear film and to quantify or estimate the total lacrimal volume involved at the ocular surface. The most common tool to evaluate the tear film stability is the fluorescein break-up time (FBUT), which measures the duration that it takes for the initially spread tear film over the cornea to break up, revealing the corneal tissue and eliciting tearing and blinking responses [11,16,17]. For the estimation of the tear volume, the most widespread tools are the tear meniscus height (TMH), phenol red and the Schirmer test [18,19]. TMH could be measured by different instruments and techniques, such as the slit lamp, multidiagnostic platforms or optic coherence tomography (OCT) instruments [11,20,21]. Researchers have found OCT to be the most reliable technique for the measurement of TMH [20]. Although slit-lamp measurements show differences from OCT, these are marginal from a clinical point of view [20]. Moreover, the values obtained with various multidiagnostic platforms that may measure the TMH are not interchangeable [21]. The Schirmer test is another standardized technique to evaluate the aqueous tear production, which consists of a wettable paper strip introduced onto the temporal bulbar conjunctiva for 5 min [22]. This technique could be performed with or without topical anesthesia, and with or without the eyes open. Its reproducibility has been shown to fluctuate throughout the day and across different days at the same time [23]. The primary disadvantage of the Schirmer test lies in its invasive nature, and the induced stimulation may potentially impact tear production [22,24]. In addition, this procedure is time-consuming, and both healthy individuals and those with dry eyes consistently report significant discomfort due to the strip. Consequently, many clinicians have opted for more comfortable alternatives that are as accurate as the Schirmer test in diagnosing DED, such as measuring TMH [25].

Tear film parameters typically display minimal variation over time or between eyes in healthy individuals. However, as the body loses control during ocular surface disease, the disruption of tear film homeostasis is mirrored by escalating changes over time or between eyes in diagnostic test values [4]. Moreover, inter-eye variability emerges as a distinctive characteristic of DED, where heightened variations in metrics across eyes can be valuable in clinical settings. Lower disparities might signify temporary effects within compensatory mechanisms [26,27]. The exploration of inter-eye variations emerges as a relevant field of study in scientific research, with the potential to reduce diagnostic costs and time. Currently, the diagnosis of DED, or closely related anomalies such as MGD, often necessitates extensive and costly test batteries [3,28,29,30]. Investigating inter-eye differences may offer a more efficient and optimized approach, providing valuable insights that could enhance the diagnostic precision and expedite the identification of these ocular disorders. Nevertheless, while previous reports have explored inter-eye differences as a diagnostic tool for DED, revealing disparities in results or even tests, none have delved into the utility of these differences for subtype detection [31,32,33,34,35]. This underscores the need for validation studies based on specific designs to assess the feasibility of this variable.

In addition to the aspects previously mentioned, while prior research has explored inter-eye differences in primary diagnostic indicators such as osmolarity, the potential of TMH for the diagnosis of DED or in classifying its subtypes has not been comprehensively investigated [3,26,27,34,35,36]. In clinical settings, TMH has demonstrated itself as a straightforward, dependable and reproducible tool for the evaluation of ADDE patients [11,37]. Additionally, this test holds the advantage of being performable with devices commonly found in clinical settings, such as a slit lamp, thereby minimizing costs and facilitating its integration into ocular examinations and daily healthcare routines. Based on this, the majority of recent studies have concentrated on establishing specific cut-off criteria for this parameter based on the employed device [11,20,37]. Beyond its utility in detecting ADDE, TMH has been proposed as an indicator of severity, capable of distinguishing between moderate–severe and mild–moderate types [3,11]. Based on the insights provided by the aforementioned literature, the present study aimed to investigate the diagnostic potential of the absolute inter-eye differences of TMH to detect the presence of ADDE.

## 2. Materials and Methods

### 2.1. Sample

A total of 260 clinical histories of patients (mean age ± standard deviation [SD] = 50.1 ± 13.97 years old, 199 female and 61 male) who attended the optometry clinic were included in the study. All of them were participants who visited the center for an ocular surface examination, referred by their medical doctor or by the health service of the institution based on dry eye complaints; this diagnostic was verified by the TFOS DEWS II criterion. In all cases, participants had no previous record of ocular surgery (including refractive surgery), ocular infection, corneal or retinal abnormalities, systemic diseases and autoimmune conditions; were not pregnant or breast-feeding; and did not wear contact lenses at the time of the study [38,39]. Furthermore, participants were instructed to refrain from using any form of artificial tears from the day preceding the study [11,40]. All participants gave written informed consent for their data to be used in research studies, and all procedures followed the Helsinki Declaration. The protocol employed in the study sessions was approved by the Bioethics Committee of the institution.

Before the study, the sample size was calculated based on the TFOS DEWS II Diagnostic Methodology report’s recommended diagnostic tests. For the sample size calculation, the software PS Power and Sample Size Calculations Version 3.1.2 (Copyright© by William D. Dupont and Walton D. Plummer) was used. The SD reported in the literature on the symptomatology is 0.15 mm [11,41]. To have 80% power for a significance level of α = 0.05 (type I error associated), to detect a clinical difference between No ADDE and ADDE participants of 0.1 mm [11,41,42], the minimum number of subjects required in each group was 54, to ensure a relative ratio control/experimental of 1:1.

Participants were randomly divided into two groups: a group to determine the optimal threshold for discrimination (study group) and a second smaller group to conduct a cross-validation process (cross-validation group). In order to avoid selection bias during the receiver operating characteristics (ROC) procedure [43], a randomization process was performed by a masked observer with the Random Sample function of the SPSS statistical software v.29.0 for Windows (SPSS Inc., Chicago, IL, USA) by setting the number of cases using the Exactly Sample Size command. The cross-validation group was determined by employing the minimum number of subjects calculated to achieve reliable results, resulting in a final group size of 60 participants [3,42,43]. Meanwhile, to ensure robust statistical reliability in determining the sensitivity and specificity of the final cut-off value obtained through ROC procedures, a larger group size of 200 participants was assigned to the study group [11,43,44].

### 2.2. Study Design and Protocol

Participants attended a routinary examination where the TFOS DEWS II Diagnostic Methodology Subcommittee criterion was employed to administer a series of tests, including the ocular surface disease index (OSDI) questionnaire, tear film osmolarity, FBUT, and corneal staining, to confirm the potential diagnosis for each participant [3,45]. Additionally, TMH was assessed at each session [11]. Both eyes of each participant were involved in all procedures [46].

### 2.3. Procedure, Diagnostic and Classification Criteria

Only participants exhibiting positive symptomatology, with at least one symptom, were included in the study, based on the criteria established by the TFOS DEWS II Diagnostic Methodology Subcommittee [3,38,45]:Participants completed the OSDI questionnaire via a self-administered online form to assess the presence of symptomatology [38,47].Tear film osmolarity was measured using a TearLab osmometer (TearLab, Escondido, CA, USA) [48]. The diagnostic cut-off values for DED diagnosis were tear osmolarity ≥ 308 mOsm/L and/or an osmolarity difference between eyes < 8 mOsm/L.FBUT and corneal staining were recorded using a Topcon^®^ (Topcon Corporation, Tokio, Japan) SL-D4 slit lamp equipped with a DC4 video camera (Topcon Corporation, Japan) and non-preserved fluorescein [11,16,17]. The diagnostic cut-off values for DED diagnosis were an FBUT ≤ 10 s and/or corneal staining (Oxford grade) ≥ 2.

Additionally, TMH was assessed at each session using a Topcon^®^ SL-D4 slit lamp equipped with a DC4 video camera under an interferometer of cool white fluorescent light (Tearscope, Keeler, Windsor, UK) complemented by ImageJ v1.54h (National Institutes of Health, Bethesda, MD, USA; http://imagej.nih.gov/ij/; accessed on 13 March 2023) [49,50,51]. Once the DED diagnosis was confirmed, participants were categorized based on the TMH captured under slit-lamp illumination as either No ADDE (TMH ≥ 0.16 mm) or ADDE (TMH < 0.16 mm) [11].

### 2.4. Statistical Analysis

The study employed the SPSS statistical software v.29.0 for Windows for data analysis, with a significance level set at *p* ≤ 0.05 for all tests. Inter-eye difference parameters (Osmolarity|OD-OS| and TMH|OD-OS|) were calculated as the absolute difference between values from both eyes [36].

An initial analysis involved determining the optimal threshold for discrimination between No ADDE and ADDE based on TMH|OD-OS| using ROC procedures on the study group [11,43,44]. Sensitivity and specificity were calculated for various threshold values, and the results were graphed on ROC curves. The discriminatory capability of the predictive model was quantified by the area under the curve (AUC), with upper and lower 95% confidence intervals (CI) provided (mean ± 1.96 × SD) [11,52]. Youden’s J statistic was used to identify the optimal numerical criterion for each ROC curve [11,43,52].

In the next step, cross-validation was conducted twice to validate the TMH|OD-OS| cut-off value obtained, demonstrating diagnostic potential (AUC, *p* ≤ 0.05). Firstly, in the study group, 80% random sampling was performed, and the TMH|OD-OS| values were transformed into a dichotomous parameter using the obtained cut-off value. The association with the initial diagnosis (TFOS DEWS II Diagnostic Methodology report) was then examined using Cramer’s V, which ranges from 0 to 1, indicating predictive ability [11]. Secondly, in the cross-validation group, a similar procedure was followed, where TMH|OD-OS| was transformed into a dichotomous parameter, and the association with the initial diagnosis was verified through Cramer’s V [11].

## 3. Results

### 3.1. Assessment of TMH|OD-OS| Cut-Off Value to Differentiate between ADDE and No ADDE Participants

The study group’s (*n* = 200, mean age ± SD = 49.8 ± 14.29 years old, 152 female and 48 male) descriptive statistics are reported in Table 1. The ROC procedures showed that TMH|OD-OS| had significant diagnostic capability to differentiate between participants’ diagnoses (AUC = 0.719 ± 0.036, *p* < 0.001, 95% CI = 0.650–0.789, Figure 1). By calculating the Youden’s index (Youden’s J statistic = 0.368), there was found a cut-off value for the TMH|OD-OS| of 0.033 mm (specificity: 77.6%; sensitivity: 59.1%) to differentiate No ADDE from ADDE participants (Figure 1).

The cross-validation analysis on a random sampling of 80% showed an association of the calculated TMH|OD-OS| cut-off value with the previously proposed diagnostic criteria of the TFOS DEWS II to differentiate between No ADDE and ADDE participants (Cramer’s V = 0.363, *p* < 0.001; Fisher exact test, *p* < 0.001; Table 2).

### 3.2. Cross-Validation of TMH|OD-OS| Cut-Off Value to Differentiate between ADDE and No ADDE Participants

The cross-validation group’s (*n* = 60, mean age ± SD = 51.1 ± 12.97 years old, 47 female and 13 male) descriptive statistics are reported in Table 3. The cross-validation analysis on a cross-validation sample showed an association of the calculated TMH|OD-OS| cut-off value with the previously proposed diagnostic criterion of the TFOS DEWS II to differentiate between No ADDE and ADDE participants (Cramer’s V = 0.354, *p* = 0.006; Fisher exact test, *p* = 0.010; Table 4).

## 4. Discussion

DED is a common ocular condition with various etiologies, and its diagnosis often relies on a combination of clinical assessments. ADDE is a subtype of dry eye characterized by reduced tear production, and accurate diagnostic tools are crucial for effective management [1,9,10]. The present study aimed to assess the diagnostic potential of the absolute inter-eye difference in TMH|OD-OS| to differentiate between participants with and without ADDE. The clinical significance of this parameter lies in its ability to obviate the need for invasive tear volume assessments, such as the Schirmer and phenol red tests. Additionally, it aligns with the TFOS DEWS II recommended approach by integrating TMH into the evaluation of ocular health. Advancements in this field may hold the prospect of optimizing clinical practices, ultimately benefiting both healthcare providers and patients by offering more effective and resource-efficient diagnostic methodologies.

The study findings suggest that TMH|OD-OS| could serve as a valuable diagnostic marker for the identification of ADDE, providing a non-invasive and quantitative measure. The present study assessed the TMH under Tearscope illumination combined with the ImageJ software (http://imagej.nih.gov/ij; accessed on 13 March 2023) for image analysis, a method that has been shown to be reliable and accurate regarding other similar procedures [11,20,37,50,51]. Additionally, the cut-off criterion employed on the initial ADDE detection based on the TFOS DEWS II principles was specifically calculated to detect ADDE participants and not only DED patients [53,54], showing sensitivity and specificity of 86.4% and 75.4%, respectively, for this purpose [11]. During the present study, the established cut-off value for the TMH|OD-OS| of 0.033 mm demonstrated reliable specificity and moderate sensitivity to detect ADDE. This finding has direct implications for clinical practice, offering a quantitative and objective measure for diagnostic decision making. Interestingly, and contrary to other clinical tests that demonstrate larger differences between eyes in DED participants [31,32,33,34,35,36], the present analysis found that participants without ADDE could be differentiated from those with ADDE by showing greater inter-eye differences. It is important to note that ADDE participants showed generally lower TMH values, while No ADDE participants had higher values despite these inter-eye differences. Other parameters, such as the osmolarity inter-eye variability, have been found to be useful for DED detection, suggesting that the higher osmolarity of the two eyes could be used in clinical practice because the lower value seems to reflect the transient effects of compensatory mechanisms [3,26,27,34,36]; however, some controversy has been found regarding this point in recent publications [35]. In addition to these results, a recent report supports the above idea, suggesting that not all diagnostic parameters suggested by the TFOS DEWS II Diagnostic Methodology report are candidates for a diagnosis based on inter-eye differences, emphasizing the need to explore each parameter individually in this regard [34].

In the present study, the cross-validation analysis supported the reproducibility of the findings and their alignment with established diagnostic criteria, reinforcing the reliability of TMH|OD-OS| in differentiating between ADDE and No ADDE. This cross-validation with an independent sample, performed in the second part of the analysis, provides additional strength to the study findings, suggesting that the identified cut-off value is not merely a chance occurrence but holds consistent diagnostic relevance across different participant groups. It is important to note that, due to the large number of participants recruited, before the randomization process, the demographic characteristics between both samples were very similar in terms of age (study group = 49.8 ± 14.29 vs. cross-validation = 51.1 ± 12.97 years old) and sex (study group = 76.0% female and 24.0% male vs. cross-validation = 78.3% female and 21.7% male). However, questioning the exclusive reliance on inter-eye differences as a singular diagnostic tool is crucial. The cross-validation analysis allowed for the determination of the true positives and negatives regarding the TFOS DEWS II criterion; the ratio of true positives to the total number of actual positives (ADDE cases) was reflected by the value of 13/35 (37.1%), whereas the number of true negatives among the participants without ADDE was computed as 24/25 (96.0%). The observed values of 37.1% and 96.0%, respectively, indicate that the cut-off value is more effective in correctly identifying participants without ADDE, which implies that the cut-off value is particularly reliable in ruling out No ADDE cases, making it a potentially useful screening tool. Whereas these findings are promising, such new parameters seem to be available for clinicians during routinary ocular surface assessment; while both the ROC and cross-validation analysis showed higher specificity (77.1 and 96.0, respectively), it should be noted that the sensitivity levels on both analyses (59.1% and 37.1%, respectively) still suggest that the parameter may not be used as a single indicator but in combination with other diagnostic criteria (e.g., the osmolarity or NIBUT, in line with TFOS DEWS II), biomarkers (e.g., Matrix Metalloproteinase 9), or even individual TMH measurements, to enhance the overall diagnostic accuracy [3,18,28,29,30,55].

Previous reports have explored the reliability of different dry eye assessments in estimating the tear volume during conditions of aqueous deficiency. Phenol red has shown specificity and sensitivity in the range of 93%–77.8% and 86%–25%, respectively, depending on the cut-off criteria employed [19,56]. Other research on ADDE-related situations, such as Sjogren’s syndrome (SS), where a meniscus evaluation clinical test has been employed, has shown the diagnostic performance of the Schirmer test (specificity 76.0% and sensitivity 42.0%) and FBUT (specificity 0.17% and sensitivity 92%) [57]. Similar results were obtained when another meniscus evaluation test was combined, such as strip meniscometry combined with tear function and ocular surface tests (specificity 58.16% and sensitivity 83.52%) [58]. In concordance with the present results, previous studies have also encountered challenges in achieving a clear balance between sensitivity and specificity for diagnosis. This finding reinforces the idea that it should be explored how TMH|OD-OS| could work in conjunction with other diagnostic criteria to enhance the overall diagnostic accuracy.

One of the key strengths of the present study lay in the design and recruitment of the sample, which included participants with previous dry eye complaints who underwent an ocular surface examination at the center, as referred by their medical doctor or the health service of the institution. It is important to note that the research employed a robust methodology and stringent diagnostic criteria, as outlined in the TFOS DEWS II principles [3,38,45], since the DED condition should be differentiated from other ocular manifestations, which encompass non-obvious disease involving ocular surface signs without related symptoms, including neurotrophic conditions with the presence of dysfunctional sensation. Additionally, it involved cases with symptoms but without demonstrable signs on the ocular surface, such as neuropathic pain [1,3].

The comprehensive diagnostic approach was meticulously implemented to address the diverse presentations of dry eye. The use of stringent diagnostic criteria enhances the results’ reliability, facilitating their translation and applicability in clinical settings. This emphasizes the validity and practical significance of the obtained findings. Moreover, future studies should explore whether TMH|OD-OS| could serve as a potential treatment target for the management of this subtype of DED. Investigating its role in treatment strategies holds the promise of more personalized and effective interventions, ultimately contributing to improved outcomes for patients with ADDE.

Despite the significant results, the present study had some limitations, such as the relatively small sample size employed in the cross-validation analysis. Moreover, although a higher prevalence of DED in women compared to men has been previously reported, the sex of the participants was not considered in the analysis [15]; therefore, future steps in this investigation should replicate the present analysis in a cohort composed exclusively of either males or females. Additionally, while the age range was not restricted to older people, all participants were of a nearly homogeneous ethnicity [14,15]. Furthermore, the clinical context, including factors such as disease severity and comorbidities, may influence the applicability of the identified cut-off value. Considering these limitations, future studies should cross-check the outcomes in a different sample diagnosed with ADDE, using alternative principles [28,29], to determine whether the reported cut-off value of TMH|OD-OS| could achieve similar specificity and sensitivity in diagnosis. These studies should consider sex differences, comorbidities such as SS [15,59], and variations in disease presentation or disease stage (such as preclinical or predisposition states where the meniscus is diminished) [1]. On the other hand, the study design should consider whether different healthcare environments (e.g., primary care, specialized clinics, or emergency departments) may present unique challenges that influence the applicability of the cut-off value. Furthermore, future longitudinal studies are needed to assess the predictive value of TMH|OD-OS| in monitoring disease progression and treatment responses over time [60,61,62].

## 5. Conclusions

In conclusion, the present study provides evidence for the diagnostic potential of TMH|OD-OS| in identifying ADDE. The established cut-off value, aligned with the TFOS DEWS II criteria, shows clinical relevance as a valuable adjunctive tool in the comprehensive assessment and classification of dry eye patients. Clinicians may benefit from proactively incorporating TMH|OD-OS| into ocular surface examinations, potentially improving cost-effectiveness and saving both economic and temporal resources for patients during routine evaluations. This targeted parameter could expedite the diagnostic process, enhancing the efficiency in identifying ADDE cases. The integration of TMH|OD-OS| into clinical practice may offer a valuable alternative for both diagnostic refinement and targeted therapeutic approaches in the management of ADDE.

## Figures and Tables

**Figure 1 jcm-13-00659-f001:**
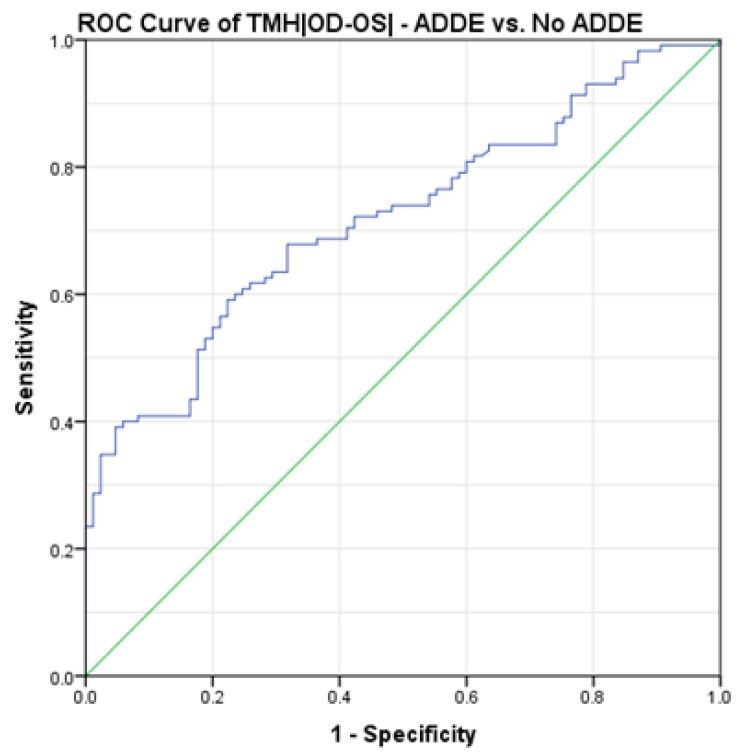
ROC curve showing the relationship between sensitivity and specificity of the TMH|OD-OS| (No ADDE vs. ADDE) according to theoretical thresholds; for each of the values observed in the study population, the sensitivity and sensibility indexes have been calculated and reported in the graph. *n* = 200. ROC = Receiver Operating Characteristic. ADDE = Aqueous Deficient Dry Eye. |OD-OS| = Inter-Eye Absolute Difference.

**Table 1 jcm-13-00659-t001:** Descriptive statistics of the study group. *n* = 200.

	Mean	SD	Minimum	Maximum
Age (years)	49.8	14.29	18.0	81.0
OSDI (Score	31.85	15.99	12.50	83.33
Osmolarity (mOsm/L)	321.9	19.31	275.0	400.0
Osmolarity |OD-OS| (mOsm/L)	11.8	12.47	0.0	71.0
Corneal Staining	0 *	1 **	0	4
FBUT (s)	7.19	6.46	0.79	65.13
TMH (mm)	0.199	0.099	0.06	0.64
TMH |OD-OS| (mm)	0.040	0.039	0.00	0.22

OD = Oculus Dexter. OS = Oculus Sinister. SD = Standard Deviation. OSDI = Ocular Surface Disease Index. FBUT = Fluorescein Break-Up Time. TMH = Tear Meniscus Height. |OD-OS| = Inter-Eye Absolute Difference. * Median for non-parametric variable. ** Interquartile range for non-parametric variable.

**Table 2 jcm-13-00659-t002:** Cross-validation analysis on a random sampling of 80% in the study group. *n* = 160.

		TMH|OD-OS| = 0.033 mm		Total
		ADDE	No ADDE	
TFOS DEWS II	ADDE	48	15	63
	No ADDE	38	59	97
	Total	86	74	160

TMH = Tear Meniscus Height. |OD-OS| = Inter-Eye Absolute Difference. ADDE = Aqueous Deficient Dry Eye. TFOS DEWS II = Tear Film and Ocular Surface Society Second Dry Eye Workshop.

**Table 3 jcm-13-00659-t003:** Descriptive statistics of the cross-validation group. *n* = 60.

	Mean	SD	Minimum	Maximum
Age (years)	51.1	12.97	21.0	76.0
OSDI (Score	30.47	14.45	12.50	77.27
Osmolarity (mOsm/L)	323.8	18.08	293.0	400.0
Osmolarity |OD-OS| (mOsm/L)	12.7	10.46	1.0	62.0
Corneal Staining	0.5 *	1 **	0	4
FBUT (s)	7.59	5.42	1.46	26.46
TMH (mm)	0.212	0.066	0.09	0.46
TMH |OD-OS| (mm)	0.034	0.035	0.00	0.19

OD = Oculus Dexter. OS = Oculus Sinister. SD = Standard Deviation. OSDI = Ocular Surface Disease Index. FBUT = Fluorescein Break-Up Time. TMH = Tear Meniscus Height. |OD-OS| = Inter-Eye Absolute Difference. * Median for non-parametric variable. ** Interquartile range for non-parametric variable.

**Table 4 jcm-13-00659-t004:** Cross-validation analysis on the cross-validation group. *n* = 60.

		TMH|OD-OS| = 0.033 mm		Total
		ADDE	No ADDE	
TFOS DEWS II	ADDE	13	1	14
	No ADDE	22	24	46
	Total	35	25	60

TMH = Tear Meniscus Height. |OD-OS| = Inter-Eye Absolute Difference. ADDE = Aqueous Deficient Dry Eye. TFOS DEWS II = Tear Film and Ocular Surface Society Second Dry Eye Workshop.

## Data Availability

Not applicable.

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
