# Peer review of "A Comprehensive Study on Tear Meniscus Height Inter-Eye Differences in Aqueous Deficient Dry Eye Diagnosis"

_jcm, 2024, doi:10.3390/jcm13030659_

Round 1

Reviewer 1 Report

Comments and Suggestions for Authors

This study investigates the diagnostic potential of the absolute inter-eye differences (|OD-OS|) of Tear Meniscus Height (TMH) for detecting Aqueous Deficient Dry Eye (ADDE) in a sample of 260 participants. Using statistical analyses, an optimal TMH|OD-OS| cut-off value was determined for diagnostic purposes. The findings indicate that the identified cut-off value of 0.033 mm has a significant diagnostic capability with reliable specificity and moderate sensitivity. The study concludes that TMH|OD-OS| is an objective tool that can enhance clinical decision-making in identifying ADDE.

Introduction

Clarify how the redefinition of DED by TFOS DEWS II aligns or diverges from previous definitions and what implications this has for diagnosis and treatment strategies.

Provide more detailed explanations or models to demonstrate how the listed characteristics of DED (tear film instability, hyperosmolarity, etc.) interact and contribute to the disease.

Address potential overlaps between ADDE and EDE, and discuss the spectrum of symptoms that might not fit neatly into either category.

Explain the methodologies used for measuring tear volume and production and discuss their accuracy, reliability, and availability in clinical settings.

Strengthen the argument for using inter-eye variability in TMH as a diagnostic tool by providing more data, validation studies, or comparisons with other diagnostic methods.

Provide a more substantial discussion on why previous studies have not utilized inter-eye differences for subtype detection and what makes this approach promising now.

Method

Question the methodology and criteria used for dividing participants into the Study and Cross-validation groups. Clarify how randomization was ensured.

Recommend a more detailed explanation or justification for the size of each group, especially why the Cross-validation group is significantly smaller.

Question the reliability and accuracy of the instruments used for assessing tear film osmolarity and TMH. How do these compare to other available methods?

Question the process for determining the optimal threshold for TMH|OD-OS| in diagnosing ADDE. Is there a risk of overfitting to the study population?

Suggest methods for further validation of the cut-off value beyond the Cross-validation Group used in the study.

Discussion

Critique the reliance on inter-eye differences as a singular diagnostic tool. Suggest exploring how this metric works in conjunction with other diagnostic criteria or biomarkers to enhance overall diagnostic accuracy.

The discussion highlights the reproducibility of findings through cross-validation. However, question the robustness of the cross-validation group given its relatively small size and recommend expanding this group in future studies for more robust validation.

Critique the practical application of the identified cut-off value, considering the clinical context and disease severity. Suggest that future research explore how different clinical settings, patient populations, or disease severities might affect the utility of the cut-off value.

Praise the use of stringent diagnostic criteria and a comprehensive approach but suggest providing more context on how these criteria compare with other commonly used standards and what additional benefits they bring to the study.

Author Response

DETAILED RESPONSES TO REVIEWER 1

Comments and Suggestions for Authors

This study investigates the diagnostic potential of the absolute inter-eye differences (|OD-OS|) of Tear Meniscus Height (TMH) for detecting Aqueous Deficient Dry Eye (ADDE) in a sample of 260 participants. Using statistical analyses, an optimal TMH|OD-OS| cut-off value was determined for diagnostic purposes. The findings indicate that the identified cut-off value of 0.033 mm has a significant diagnostic capability with reliable specificity and moderate sensitivity.

The study concludes that TMH|OD-OS| is an objective tool that can enhance clinical decision-making in identifying ADDE.

- Response: The authors would like to thank the reviewer for their detailed comments and suggestions about the manuscript. The authors believe that the comments have identified important areas that require improvement. Following the reviewers' indications, the entire manuscript has been re-edited to improve readability and clarify any unclear points. As the reviewer suggests, some changes were made in all sections. The authors agree with the referee on the need for an additional classification of sample and cross-validation analyses, as well as the need for more context at some points of the manuscript to make a rational explanation for the study.

Introduction

  1. Clarify how the redefinition of DED by TFOS DEWS II aligns or diverges from previous definitions and what implications this has for diagnosis and treatment strategies.

- Response: Thank you for pointing that out. The introduction section has now been updated to provide additional information on the redefinition of Dry Eye Disease (DED). This expansion sheds light on the implications for diagnosis and treatment strategies, offering a more comprehensive overview.

  1. Provide more detailed explanations or models to demonstrate how the listed characteristics of DED (tear film instability, hyperosmolarity, etc.) interact and contribute to the disease.

- Response: Thank you for the indication, detailed explanations about the vicious circle of the disease have been implemented in the introduction section.

  1. Address potential overlaps between ADDE and EDE, and discuss the spectrum of symptoms that might not fit neatly into either category.

- Response: Thank you for the suggestion. The introduction section has been revised to incorporate a brief explanation of how ADDE and EDE outcomes might overlap, particularly in older age and severe stages of the disease. This addition aims to recognize potential intricacies in symptomatology that may not neatly fall into either category, contributing to a more detailed understanding of the spectrum of Dry Eye Disease manifestations.

  1. Explain the methodologies used for measuring tear volume and production and discuss their accuracy, reliability, and availability in clinical settings.

- Response: Thank you for the commentary. The manuscript has been updated to include additional details on the methodologies employed to measure tear dynamics.

  1. Strengthen the argument for using inter-eye variability in TMH as a diagnostic tool by providing more data, validation studies, or comparisons with other diagnostic methods.

- Response: Thank you for the recommendation. In response to the suggestions in this and the following question, the concluding section of the introduction has been expanded to establish a comprehensive rationale for the current study, drawing on previous findings and the identified gaps in research. However, should the reviewers find it necessary, further enhancement of this data could be considered in future revisions.

  1. Provide a more substantial discussion on why previous studies have not utilized inter-eye differences for subtype detection and what makes this approach promising now.

- Response: Thank you for the suggestion. The manuscript has been updated to include an expanded rationale for the study. This can be found at the end of the introduction section, adjacent to the study's objective.

Method

  1. Question the methodology and criteria used for dividing participants into the Study and Cross-validation groups. Clarify how randomization was ensured.

- Response: Thank you for the indication. Randomization was carried out using the SPSS statistical software v.29.0 for Windows (SPSS Inc., Chicago, United States). References to the software and commands used have been included in the text. Furthermore, the "2.1 Sample" section has been slightly reorganized to better align with the current and subsequent questions. The authors express their gratitude to the reviewer for highlighting the need for a more thorough explanation of the sample size design in the previous version, an aspect they acknowledge and have now addressed.

  1. Recommend a more detailed explanation or justification for the size of each group, especially why the Cross-validation group is significantly smaller.

- Response: Thank you for the feedback. The requested explanation has been expanded, providing a more detailed justification for the size of each group. Additionally, in order to adhere to the indications provided in the current and previous questions, the "2.1 Sample" section has been restructured.

  1. Question the reliability and accuracy of the instruments used for assessing tear film osmolarity and TMH. How do these compare to other available methods?

- Response: Thank you for the suggestion. Given that the study is focused primarily on the TMH, specific data regarding the specificity, sensitivity, and repeatability of the measurement employed method has been incorporated into the discussion section. However, no new data or references have been introduced regarding the osmolarity or the other diagnostic tests used (OSDI, FBUT, or corneal staining) under the TFOS DEWS II principles. The authors agree with the reviewer's observation that tests such as osmolarity have exhibited high variability, and their utility has been even questioned (Cornea, Baenninger et al., 2018; Cont Lens Anterior Eye, Szczesna-Iskander et al. 2016; JAMA, Bunya et al. 2015; Clin Exp Optom, Pena-Verdeal et al, 2023). However, understanding that this may extend beyond the manuscript's scope and potentially impact readability and the manuscript's core message, the authors have chosen not to extensively discuss this topic. Nevertheless, if the reviewers believe, in future revisions, that a more detailed discussion of this issue is warranted, it can be added to the discussion section.

  1. Question the process for determining the optimal threshold for TMH|OD-OS| in diagnosing ADDE. Is there a risk of overfitting to the study population?

- Response: In order to achieve robust results in determining the optimal threshold, a substantial initial sample size was employed before subjecting it to a rigorous randomization process, which was later verified by a cross-validation process (Phys Med Biol., Obuchowski et al., 2008). The literature emphasizes the substantial impact of sample size on ROC procedures, with more reliable outcomes observed when the sample size meets or exceeds the minimum requirement. Measures were taken to prevent verification or selection bias during recruitment, based on a prior randomization process before analysis. The derived values were subsequently validated within both the initial sample and a second sample that was not included in, nor affected by the ROC calculation. Minor adjustments have been made to the text to reflect these statistical principles, however, if the reviewer deems it necessary, a long explanation regarding this issue could be added to the text in future revisions.

  1. Suggest methods for further validation of the cut-off value beyond the Cross-validation Group used in the study.

- Response: An additional cross-validation analysis was included in the initial sample, supplementing the previous one conducted in the Cross-validation Group. Furthermore, the discussion section now includes a recommendation for future studies to assess the robustness of the current cut-off value under varied conditions, along with the recommendation to perform longitudinal studies in both, the diagnostic and treatment response fields, to further cross-validate the present findings.

Discussion

  1. Critique the reliance on inter-eye differences as a singular diagnostic tool. Suggest exploring how this metric works in conjunction with other diagnostic criteria or biomarkers to enhance overall diagnostic accuracy.

- Response: Thank you for the suggestion. This concern has been addressed by incorporating it into the discussion at different points, highlighting the ongoing importance of combining various methods during ocular surface anomaly diagnosis.

  1. The discussion highlights the reproducibility of findings through cross-validation. However, question the robustness of the cross-validation group given its relatively small size and recommend expanding this group in future studies for more robust validation.

- Response: The reviewer is right. Following the guidance provided by both reviewers, improvements have been made throughout the entire manuscript regarding the cross-validation process and the study's limitations to enhance clarity on these matters. However, if the reviewers deem it necessary, the issue could be enlarged or modified in future reviews.

  1. Critique the practical application of the identified cut-off value, considering the clinical context and disease severity. Suggest that future research explore how different clinical settings, patient populations, or disease severities might affect the utility of the cut-off value.

- Response: Thank you for your suggestion. A paragraph outlining future directions and recommendations has been incorporated into the manuscript's discussion section following the elaboration and enhancement of the study limitations.

  1. Praise the use of stringent diagnostic criteria and a comprehensive approach but suggest providing more context on how these criteria compare with other commonly used standards and what additional benefits they bring to the study.

- Response: The raised concern has been addressed by integrating a specific paragraph discussing the specificity and sensitivity of previously related tear volume tests. Furthermore, the discussion on this matter has been improved throughout the entire manuscript, underscoring the importance of inter-eye differences as a singular diagnostic tool. The authors express their sincere gratitude to the reviewer, as they believe that these modifications significantly enhance the scientific impact of the present study results.

Reviewer 2 Report

Comments and Suggestions for Authors

Dry eye disease is a condition characterized by reduced eye lubrication by the tears. A clinical study by Pena-Verdeal et al developed a diagnostic tool to identify aqueous deficient dry eye. The study is interesting and has clinical significance. The authors described the article briefly with clarity. The experiment was designed well with strong methods. The authors should address these queries before final decision.

1.     Why the gender of the participants not considered in the analysis

2.     Does any participant have any other eye disorders such as retinal/corneal dysfunctions. If yes please include that is the analysis

3.     The clinical relevance of the study should highlight more in the conclusion.

Author Response

DETAILED RESPONSES TO REVIEWER 2

Comments and Suggestions for Authors

Dry eye disease is a condition characterized by reduced eye lubrication by the tears. A clinical study by Pena-Verdeal et al developed a diagnostic tool to identify aqueous deficient dry eye. The study is interesting and has clinical significance. The authors described the article briefly with clarity. The experiment was designed well with strong methods. The authors should address these queries before final decision.

- Response: The authors want to thank the reviewer for the effort towards improving the present manuscript. The authors have answered all comments from the two reviewers, and hope that the editor and reviewers will find the paper suitable for publication; however, if the reviewer deems it necessary, more explanations could be enlarged in further revision. Please, find enclosed below the responses to the specific comments.

  1. Why the gender of the participants not considered in the analysis

- Response: Thank you for your comments. The authors agree with the reviewer on the importance of analyzing gender differences, such as the higher prevalence of DED in women compared to men, primarily due to hormonal and hereditary variations (Ocul Surf. Sullivan et al., 2017). Consequently, the majority of patients attending our center are women (approximately 65-75% of them), as reflected in the updated descriptive data provided in the manuscript. This inherent characteristic of the pathology poses challenges in conducting well-balanced gender studies. The limitation and future direction paragraphs of the manuscript were enlarged, including a limitation related to this issue that has been added along with new clarifications in the introduction, results, and discussion.

  1. Does any participant have any other eye disorders such as retinal/corneal dysfunctions. If yes please include that is the analysis

- Response: The reviewer is right; the exclusion/inclusion information was not included in the present version of the manuscript. Details regarding to the ocular and general disorders considered in the selection process have been incorporated into the "2.1. Sample" section.

  1. The clinical relevance of the study should highlight more in the conclusion.

- Response: Thank you for the suggestion. The conclusion was expanded and the clinical implications of the present result were added to the text.

Round 2

Reviewer 1 Report

Comments and Suggestions for Authors

The authors solved all the comment correctly

Author Response

The authors solved all the comment correctly

- Response: The authors wish to extend their gratitude once again to the reviewer for diligently reviewing the current document. The authors acknowledge that the comments have pinpointed crucial areas that necessitate improvement. As a result, the manuscript has been further refined to delve more deeply into the examined issue.
